# Serum GFAP and NfL augment a metabolomics-driven strategy for long-term prediction of multiple sclerosis progression

## Abstract

**Background** Reliable biomarkers for predicting disease progression in multiple sclerosis (MS) are crucial for advancing precision medicine and optimising treatment strategies. This study evaluates the predictive potential of serum nuclear magnetic resonance (NMR)-based metabolomics, individually and in combination with well-established biomarkers of neuroinflammation (serum glial fibrillary acidic protein, sGFAP) and axonal damage (neurofilament light chain, sNfL), in an extreme-phenotype subset of the Swiss Multiple Sclerosis Cohort (SMSC).

**Methods** Serum samples were analysed using NMR-based metabolomics, along with quantification of sNfL and sGFAP. Supervised multivariate analysis was performed to differentiate MS phenotypes and identify future progressors. Multivariable receiver operating characteristic (ROC) analysis evaluated predictive performance, with key metabolite findings validated in an independent Oxford MS cohort.

**Results** NMR-based metabolomics reliably distinguishes relapsing-remitting MS (RRMS) from secondary-progressive MS (SPMS) and predicts individual transitions. The identified predictive metabolites (lipoproteins, glutamine, alanine, valine, glucose) are also associated with progression independent of relapse activity (PIRA), a clinically relevant marker of sustained disability worsening. This demonstrates that the approach can both stage disease and forecast progression irrespective of stage. ROC analysis shows strong predictive performance (AUC = 0.81, $p$ = 0.001), with external validation confirming robustness. Integration of NMR-metabolomics with sGFAP and sNfL further improves accuracy, yielding AUCs of 0.91 ($p$ < 0.0001) and 0.87 ($p$ = 0.0002), respectively, supported by independent validation.

**Conclusions** The integration of metabolic and protein biomarkers enables both accurate staging of RRMS versus SPMS and, critically, early prediction of progression irrespective of stage. This dual capability provides a clinically actionable, serum-based tool that can refine monitoring, improve therapeutic decision-making, and support a shift towards stage-agnostic, progression-focused care in MS.

## Plain Language Summary

Predicting how multiple sclerosis (MS) will develop is challenging but essential for guiding treatment. Disability can worsen even in the absence of relapses, and current assessment tools provide limited ability to anticipate this. In this study, we analysed blood samples from people with MS using metabolomics, a method that profiles many small molecules reflecting energy and lipid metabolism. We combined this information with two established blood proteins that indicate nerve damage and inflammation. Together, these measures distinguished between different stages of disease and identified individuals at greater risk of progression. This approach points towards the development of a simple blood test that could support more personalised care through earlier and better-informed treatment decisions.

Historically, patients with multiple sclerosis (MS) have been dichotomised into those in the relapsing-remitting MS (RRMS) stage or those who have transitioned to the secondary progressive MS (SPMS) stage[1,2]. Currently, this classification relies heavily on clinical assessment, typically through the expanded disability status scale (EDSS)[3], in conjunction with magnetic resonance imaging (MRI)[4]. However, despite the necessity for distinct treatment strategies at each stage, a standardised method for identifying the transition from RRMS to SPMS, and of subclinical progression per se, remains elusive[1–3]. This challenge likely stems from the somewhat arbitrary nature of the RR/SP classification, as some

✉e-mail: daniel.anthony@pharm.ox.ac.uk

individuals exhibit stable disability while others experience deterioration independent of RR/SP staging[2,5,6]. Moreover, an increasing body of evidence highlights the limitations of this dichotomy, as progression independent of relapse activity (PIRA) is frequently observed already during the RR phase, underscoring the need to move away from phenotypic classifications and towards a focus on underlying biological factors driving disease progression. However, the absence of validated biomarkers of progression[2,7] and the variability of treatment responses complicate this endeavour[1]. Since subclinical neurodegenerative processes begin early in MS, relying on clinical markers of disease progression falls short to anticipate the future disease course and the need for timely therapeutic intervention in the concept of precision medicine. Therefore, identifying biomarkers capable of staging or predicting the rate of progression would offer valuable personalised prognostic insights, enabling timely and optimised treatment strategies.

Serum glial fibrillary acidic protein (sGFAP) is an established biomarker for astrocyte injury and degenerative neurological disorders[8–12], including MS. A marked disparity in sGFAP concentrations was observed between SPMS and RRMS patients, with SPMS individuals demonstrating higher levels[9,13,14]. Moreover, recent studies have proposed a correlation between sGFAP abundance and disease progression[13–19]. Conversely, concentrations of serum neurofilament light chain (sNfL) reflect neuroinflammatory activity[20–23], neuroaxonal damage due to ongoing brain inflammation[22,24,25], and responsiveness to treatment[21,26–29]. Elevated sNfL was found to be linked to relapse incidence in RRMS[11,21,30–33], and to lesser degree with disease progression[16,21].

We have previously demonstrated, that serum nuclear magnetic resonance (NMR)-metabolomics effectively discriminates between RRMS and SPMS[34–37] and identifies relapse activity in RRMS patients[38]. While NMR holds promise in identifying biomarkers for MS clinical stages[34–36,39,40], studies utilising NMR-metabolomics to monitor MS progression are lacking[39,41]. Additionally, the lack of intraindividual longitudinal studies across different disease stages represents a critical gap, limiting the translation of findings into personalised medical application.

The Basel extreme phenotype cohort[16] provides a valuable platform for identifying progression biomarkers, with the potential for validation in larger and more heterogeneous populations. Despite its modest size, its long-term follow-up enables detailed assessment of biomarker dynamics and the RRMS-SPMS transition. In this study, we shift focus from stage-based classification towards defining molecular signatures of progression itself. We confirm that NMR-detectable serum metabolites distinguish RRMS from SPMS and show that baseline metabolic signatures predict future progression. Moreover, we demonstrate that integrating NMR metabolites with sGFAP and sNfL enhances the prediction of progression, establishing a molecular profile of "progressors" that transcends conventional staging.

## Methods
### Study subjects and procedures
This study utilised clinical samples from the longitudinal Swiss Multiple Sclerosis Cohort (SMSC), a consortium of tertiary referral hospitals across Switzerland. The SMSC is registered with ClinicalTrials.gov (NCT02433028). Ethical approval was obtained from independent ethics committees at each participating centre (Aarau, Basel, Bern, Geneva, Lausanne, Lugano, and St Gallen), with primary approval granted by the Ethikkommission Nordwest- und Zentralschweiz (EKNZ; approval number EKNZ 48/12). All participants provided written informed consent. The study adhered to STROBE reporting guidelines. Patients with MS were included based on matched baseline EDSS scores, stability or worsening disability, and absence of relapses throughout follow-up. Non-progressors experienced no PIRA events, while progressors reported ≥1 PIRA event, defined as confirmed disability worsening (CDW) without relapse activity. CDW was classified as an EDSS increase of ≥1.5 points (baseline EDSS = 0), ≥1.0 points (EDSS 1.0–5.5), or ≥0.5 points (EDSS ≥ 6.0), confirmed after ≥6 months[16].

The validation cohort was derived from the Oxford METabolomics Multiple Sclerosis (MET) cohort[36], with patients recruited from the John Radcliffe Hospital, part of the Oxford University Hospitals NHS Foundation Trust. Ethical approval was obtained through the Oxford Radcliffe Biobank and the NRES Committee South Central-Oxford C (REC reference: 09/H0606/5 + 5). All participants provided written informed consent.

Progressor and non-progressor subsets were selected based on extreme ambulatory trajectories, with groups matched for baseline EDSS. Non-progressors exhibited stable ambulatory function, maintaining unrestricted walking ability at both baseline and 2-year follow-up, whereas progressors experienced pronounced ambulatory decline, defined by a reduction in walking capacity to ≤500 meters. Based on these criteria, 19 progressors and 24 non-progressors were identified. Compared with the Basel discovery cohort, the Oxford cohort encompassed a broader clinical spectrum, providing a complementary and more generalisable setting for validation.

### sGFAP and sNfL measurements
Blood samples were stored at −80 °C following standardised procedures[42]. Concentrations of sGFAP and sNfL were measured, as part of a previous study, in duplicate using ultrasensitive single molecule array (Simoa) technology (Quanterix, Inc., USA) the singleplex Simoa GFAP Discovery Kit (Cat. 102336)[16] and Simoa Nf-Light Kit (Cat. 104364)[21], respectively.

### NMR sample preparation
Samples for $^1$H NMR metabolomics analysis were prepared following a well-established procedure[35,38]. Serum samples were thawed at room temperature and 150 μL was combined with 400 μL of 75 mM sodium phosphate buffer (in $D_2O$, pH 7.4), then transferred to 5 mm NMR tubes (Norell™, Merck, UK).

### $^1$H NMR metabolomics data acquisition and processing
Global (untargeted) metabolomics encompasses the detection of all discernible metabolites within a given sample. All $^1$H NMR metabolomics experiments were performed using a 700-MHz Bruker AVII spectrometer operating at 16.4 T equipped with a $^1$H ($^{13}$C/$^{15}$N) TCI cryoprobe (Department of Chemistry, University of Oxford), as described previously[35,36,38,43]. $^1$H NMR spectra were acquired using a 1D NOESY presaturation scheme and a spin-echo Carr-Purcell-Meiboom-Gill (CPMG) sequence. CPMG spectra were manually phased, baseline corrected, and chemical shifts referenced to lactate -$CH_3$ (δ = 1.33 ppm) in Topspin 4.1.4 (Bruker, Germany). Next, in ACD/NMR processor academic edition 12.01 (Advanced Chemistry Development, Inc.), the 0.50 − 4.20 ppm and 5.20 − 8.50 ppm regions were divided into 0.02 ppm width bins and the sum-normalised integrals of each bin were exported for multivariate statistical analysis. The metabolite assignment was performed with reference to literature[35,38,41,44] and the Human Metabolome Database[45]. Additional confirmation was attained through the inspection of 1D total correlation spectroscopy (TOCSY) spectra.

### Statistical analysis
Multivariate analysis was performed in R software 4.2.1 (R foundation for statistical computing, Vienna, Austria) using in-house R scripts and the *ropls* package[46]. Prior to analysis, all datasets were z-scored, with sGFAP and sNfL z-scored while adjusting for age and BMI[21]. "Orthogonal partial least square discriminant analysis" (OPLS-DA) was performed, employing 10-fold cross-validation in combination with repetition and permutation testing, as previously described[41,43]. Biomarker selection was based on variable importance in projection (VIP) scores derived from the OPLS-DA model, with the most discriminatory features identified at the inflection point of the VIP score distribution[41,43]. The analysis focused initially on the clinical stratification into RRMS and SPMS. A subsequent multivariate model was constructed using a subset of non-transitioning RRMS patients ($N = 10$) and SPMS participants who were clinically diagnosed as SPMS at the time of baseline sampling ($N = 9$). Patients who transitioned from RRMS to SPMS during the study ($N = 8$) and an additional cohort of stable RRMS

patients ($N = 10$) were excluded from the model as an independent validation set for leave-one-out cross-validation.

To validate our key findings, we used an independent Random Forest (RF) model with 500 trees, constructed using the *randomForest* R package[47], to confirm the OPLS-DA-derived outcomes. A 10-fold cross-validation was applied and repeated 100 times. Feature selection was performed using *t*-tests, receiver operating characteristic (ROC) curve, area under the curve (AUC), and Cohen's D effect size. Model accuracy, sensitivity, specificity, and AUC were assessed and averaged across folds and repetitions, and permutation testing was performed to generate random classification baselines. The VIP scores and out-of-bag error rates were determined for the constructed model.

Predictor variables identified through VIP scores from OPLS-DA and RF models were evaluated for their diagnostic potential. Univariate data analysis was performed in GraphPad Prism 10 software. Student's *t*-test ($p < 0.05$) was used to compare metabolite concentrations between progressors and non-progressors. Quantitative data are shown as Tukey box-plots with IQR and whiskers ($1.5\times$ IQR). Benjamini-Hochberg FDR correction was applied where appropriate. The Pearson correlation analysis was conducted using the *corrplot* R package. The Kaplan-Meier curves utilised baseline levels of the identified discriminatory metabolites, standardised as *z*-scores, and log-rank (Mantel-Cox) tests were used to compare groups. The "event" plotted on the curve was defined as the time point when the patient was clinically classified as SPMS or experienced an EDSS increase of $\geq 1.5$ units (for EDSS > 6, a 0.5-unit increment was counted as 1 unit) relative to baseline. The identified metabolites were evaluated using ROC curves, AUC, and optimal thresholds calculated with the *pROC* R package[48]. The AUC threshold was determined using the Youden Index, maximising the sum of sensitivity and specificity on the ROC curve. To capture complex relationships, two-way interaction terms between predictor variables were evaluated. A logistic regression model was then fitted to classify the binary outcome (progressors vs non-progressors). Given the limited sample size and the rigorous prior cross-validation of metabolites using OPLS-DA and RF the ROC curves were generated without additional cross-validation. To minimise overfitting, the final models were restricted to 5-6 predictors, yielding a feature-to-sample ratio of approximately 1:6-1:7 in the discovery cohort. The DeLong test was applied to assess the model discriminatory power, calculating the *p*-value for the ROC AUC to test the null hypothesis of no discrimination (AUC = 0.5).

## Results
### Patient characteristics
The cohort comprised 18 progressors ($\geq 1$ PIRA event) and 19 non-progressors (absence of PIRA events). In both the SMSC discovery (Table 1) and Oxford validation cohort (SI Table 6), progressors and non-progressors were closely matched for baseline EDSS and age, with no significant demographic differences apart from those inherently associated with progression. As expected, progressors exhibited significantly higher EDSS scores at the last sample ($p < 0.001$), greater number of PIRA events ($p < 0.001$), and a markedly higher prevalence of SPMS (SPMS; 44.4% vs 5.0%; $p < 0.001$) and transition from RRMS to SPMS (44.4% vs 0.0%; $p < 0.001$). Treatment patterns differed between groups. At baseline, progressors were more frequently treated with monoclonal antibodies, while non-progressors predominantly received oral therapies. By the last sample, monoclonal antibody use increased among progressors, whereas non-progressors largely remained on oral therapies.

### Monitoring the transition from RRMS to SPMS at the individual level using NMR serum metabolomics
We first confirmed metabolome stability in patients who had a consistent diagnosis of RRMS ($N = 20$) or SPMS ($N = 9$) throughout the follow-up period (Supplementary Table 1). Our findings revealed a significant increase in high-density lipoproteins (HDLs, $\delta$ 0.86–0.88 ppm; region dominated by HDLs mobile -$CH_3$; $p = 0.003$) and low-density lipoproteins (LDLs, $\delta$ 1.26–1.28 ppm; region dominated by LDLs mobile (-$CH_2$-)$_n$; $p = 0.023$),

along with a significant decrease in *N*-acetylated glycoprotein (GlycA, $\delta$ 2.04–2.06 ppm region overlapping with mobile lipoprotein =CH-$CH_2$-$CH_2$- resonances, $p = 0.014$) and glucose ($\delta$ 3.76–3.78 ppm; $p = 0.032$) in RRMS compared to SPMS, consistent with findings from our previous work[34–37].

To assess metabolome ability to capture the transition from RRMS to SPMS, we constructed an OPLS-DA model using the 9 SPMS individuals and a subset of 10 stable RRMS individuals who did not transition during the follow-up period (Supplementary Table 2). The model achieved an accuracy of $74.6 \pm 2.3\%$, sensitivity of $73.9 \pm 3.1\%$, specificity of $77.1 \pm 6.2\%$ (Supplementary Fig. 1). Compared to previously reported studies focusing on dichotomising RRMS and SPMS[34–36], glutamine ($\delta$ 2.12–2.2.14 ppm; $p = 0.0002$), alanine ($\delta$ 1.48–1.50 ppm; $p = 0.006$), and valine ($\delta$ 0.98–1.00 ppm; $p = 0.021$), all of which were elevated in SPMS participants, emerged as highly discriminatory metabolites (Supplementary Table 3). To corroborate the OPLS-DA-derived findings, we applied a RF algorithm to the same dataset, resulting in comparable performance metrics: an accuracy of $74.9 \pm 2.6\%$, sensitivity of $74.4 \pm 4.0\%$, and specificity of $76.7 \pm 3.4\%$. Most discriminatory variables identified by OPLS-DA model also emerged as significant in the RF analysis, with the exception of glutamine and GlycA (Supplementary Table 4). This minor discrepancy suggests that glutamine and GlycA may have reduced relative importance in RF due to weaker classification contributions or redundancy with other metabolites. However, the consistency of key features across methods confirms model robustness.

The OPLS-DA model was used to predict the classification of samples from eight individuals who transitioned from RRMS to SPMS during follow-up, as determined by clinical examination (for demographic details, see Supplementary Table 5). As part of the validation process, the model predictive accuracy was tested on a separate cohort of 10 stable RRMS patients, which had been excluded from the initial model training (Supplementary Fig. 2). Additionally, leave-one-out cross-validation was performed on the remaining individuals used to train the multivariate model, consisting of 10 RRMS and 9 SPMS cases (Supplementary Figs. 3 and 4). The subsequent analysis of predictive component scores revealed that transitioning patients exhibited significantly greater metabolomic fluctuations (Fig. 1, cases 1–8) compared to stable RRMS ($N = 20$, Supplementary Fig. 2, 3) and non-transitioning SPMS ($N = 9$, Supplementary Fig. 4) individuals. Building on these results, we examined the alignment between conventional clinical staging and NMR metabolomics (Fig. 1, column 1; "Metabolomics"). Focusing on the pivotal transition point to SPMS, we observed perfect agreement between clinical and metabolomics-based diagnoses for two individuals (Fig. 1, cases 2 and 3). Conversely, for two individuals, the metabolomics-based model indicated an earlier transition than captured by the clinical diagnosis (Fig. 1, cases 1 and 7), while for four individuals, a later transition was indicated by metabolomics (Fig. 1, cases 4, 5, 6, 8). Analysing the general trend, as represented by the fluctuations in individual metabolomes, revealed overall agreement for five participants (Fig. 1, cases 1, 2, 4, 6, 8). In contrast, for two participants, the metabolome initially reached an "SPMS-apex" and gradually reverted to an RRMS-like composition, a pattern not reflected in the clinical assessment (Fig. 1, cases 3, 5). For one participant, the metabolomic profile suggested SPMS at the first sample (Fig. 1, case 7), while the clinical diagnosis indicated the transition at a later time point. Examining the EDSS scores associated with each sample did not reveal any direct correlation with the transition, in line with its known insensitivity to change.

Given recent progress in utilising sGFAP levels to monitor MS progression, we next assessed their agreement with clinical staging (using *z*-scores of sGFAP concentration; Fig. 1, column 2; "sGFAP"). The metabolomics-based model demonstrated a higher degree of agreement with the clinical diagnosis compared to the sGFAP-based model. The agreement (within the 95% CI) with the metabolomics-based model allowed for a consistent classification of 19/27 RRMS samples (70%) and 47/64 SPMS samples (73%). Contrarily, for sGFAP alone, the overall agreement with the clinical examination-based diagnosis was observed for 15/27 RRMS

**Table 1 | Demographic characteristics of progressing (*N* = 18) and non-progressing (*N* = 19) MS individuals (for more details on patient selection criteria, please refer to the original study[16])**

| Parameter | Progressors | Non-progressors | p-value |
|---|---|---|---|
| *N* | 18 | 19 | |
| Female (%) | 11 (61.1%) | 12 (63.2%) | 0.99 |
| Age first sample | 46.8 ± 10.5 | 45.1 ± 9.7 | 0.63 |
| Age last sample | 52.9 ± 10.9 | 52.1 ± 9.9 | 0.82 |
| Number of visits | 10.6 ± 2.8 | 8.9 ± 1.6 | 0.13 |
| Length of follow-up (months) | 79.8 ± 20.8 | 82.4 ± 16.0 | 0.41 |
| BMI (baseline) | 25.3 ± 4.6 | 26.7 ± 4.7 | 0.40 |
| EDSS first sample | 3.9 ± 1.2 | 3.3 ± 1.2 | 0.11 |
| EDSS last sample | 6.3 ± 1.1 | 2.9 ± 1.5 | 6.40E-9 |
| T2w lesion volume, median (IQR), mL (baseline) | 16.3 (12.8–44.7) | 10.9 (2.7–19.7) | 0.21 |
| **No. of PIRA events** | | | |
| 0 | 0 (0) | 19 (100) | 5.7E-11 |
| 1 | 6 (33.3) | 0 (0) | |
| 2 | 8 (44.4) | 0 (0) | |
| 3 | 4 (22.2) | 0 (0) | |
| **Disease course** | | | |
| †RRMS | 2 (16.6%) | 18 (95.0%) | 1.1E-6 |
| †SPMS | 8 (44.4%) | 1 (5.0%) | |
| ††Transitioning RRMS → SPMS | 8 (44.4%) | 0 (0.0%) | |
| **Therapy - baseline** | | | |
| Untreated | (7) | (3) | |
| Orals | fingolimod (4) | fingolimod (5) | |
| | dimethyl fumarate (1) | dimethyl fumarate (1) | |
| | azathioprine (1) | — | |
| Monoclonal antibodies | ocrelizumab (2) | — | |
| | natalizumab (2) | natalizumab (5) | |
| | rituximab (1) | — | |
| Platform therapies | — | glatiramer acetate (1) | |
| | — | interferon beta 1a (1) | |
| | — | interferon beta 1b (3) | |
| **Therapy – last sample** | | | |
| Untreated | (4) | (1) | |
| Orals | (0) | fingolimod (8) | |
| | — | dimethyl fumarate (1) | |
| | — | ozanimod (1) | |
| | — | teriflunomide (1) | |
| Monoclonal antibodies | ocrelizumab (8) | ocrelizumab (1) | |
| Platform therapies | natalizumab (6) | natalizumab (2) | |
| | (0) | interferon beta 1a (1) | |
| | — | interferon beta 1b (2) | |
| | — | peginterferon beta 1a (1) | |

Data are presented as mean ± standard error of the mean (SEM). Continuous variables were compared using Student's two-sided t-test (or Wilcoxon rank-sum test when data were not normally distributed). Categorical variables were compared using Fisher's exact test. † Diagnosis remained stable from baseline to last follow-up. †† Diagnosis changed between baseline to follow-up. No adjustment for multiple comparisons was applied in Table 1 in order to maximise sensitivity.

samples (56%) and 45/64 SPMS samples (70%). Finally, we assessed whether the combination of NMR metabolomics and sGFAP yields improved agreement with the clinically defined diagnosis by incorporating sGFAP concentration into the OPLS-DA model (Supplementary Fig. 5). Integrating metabolomics data with sGFAP concentrations (Fig. 1, column 3; "Metabolomics + sGFAP") improved sample classification accuracy for RRMS and SPMS compared to metabolomics-only or sGFAP-only models, achieving agreement with clinical diagnosis in 22/27 (82%) RRMS and 57/64 (84%) SPMS samples.

### The baseline NMR serum metabolome is an indicator of future transition to SPMS

Having demonstrated that the identified metabolites reflect disease stage at a given time point, we next evaluated the potential of baseline metabolites to predict the transition from RRMS to SPMS. Survival analysis was conducted using baseline samples from all stable RRMS (*N* = 20) and transitioning (*N* = 8) individuals. The "event" in the Kaplan-Meier curves was defined as the transition to SPMS (Supplementary Fig. 6). The results indicated that the

baseline metabolome partially predicted the RRMS-to-SPMS transition. Among the metabolites, lipoproteins emerged as the most predictive biomarkers (*p* = 0.045; Supplementary Fig. 6a), with lower baseline lipoprotein levels associated with a higher likelihood of transitioning to SPMS. In contrast, baseline levels of alanine (*p* = 0.171; Supplementary Fig. 6b), glutamine (*p* = 0.183; Supplementary Fig. 6c), valine (*p* = 0.234; Supplementary Fig. 6d), glucose (*p* = 0.392; Supplementary Fig. 6e), and GlycA (*p* = 0.303; Supplementary Fig. 6f) were not significantly associated with future transition to SPMS. Nevertheless, several metabolites demonstrated consistent trends, and when analysed jointly in multivariable models, they are likely to represent transition dynamics more robustly than individual metabolites alone (Fig. 1).

### The baseline NMR serum metabolome prognosticates rate of progression independently of actual clinical disease stage

Expanding on the capacity of baseline metabolite concentrations to forecast disease trajectory and the transition to SPMS, we aimed to ascertain whether the relative concentration of the discriminatory serum metabolites (HDLs,

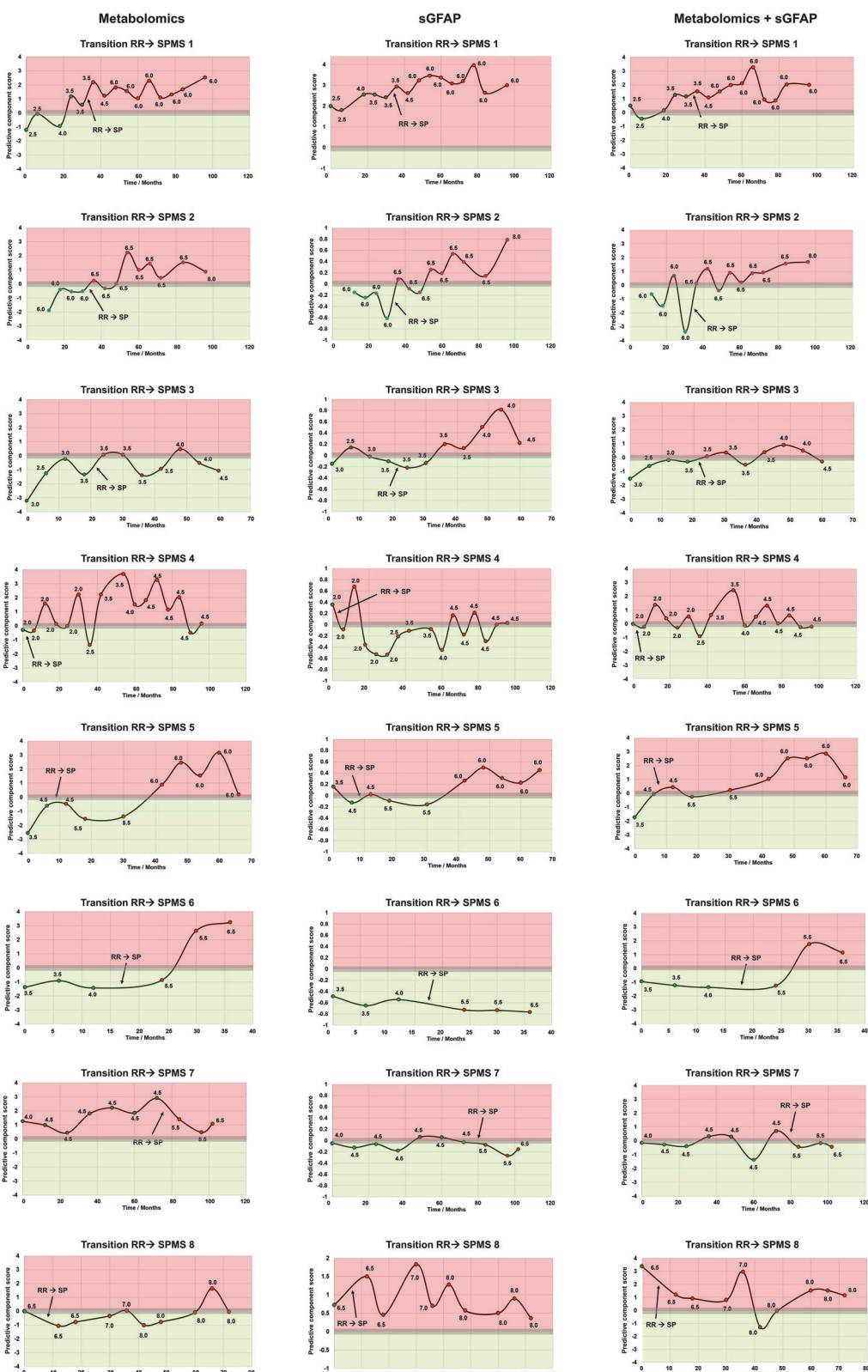

**Fig. 1 | Serum metabolomics as an alternative approach to stage MS.** Predicted OPLS-DA scores of eight individuals undergoing the transition from RRMS to SPMS are illustrated. The red-shaded area represents the SPMS-like metabolome, while the green area corresponds to the RRMS-like metabolome, the grey area represents 95% CI. Each datapoint corresponds to a specific time point when the blood sample was collected, with red data points indicating clinical SPMS diagnosis and green points clinical RRMS diagnosis. Each datapoint is labelled with the corresponding EDSS score. The arrows emphasise the transition from RRMS to SPMS in each individual, according to the clinical examination results. Column 1 - Metabolomics-based multivariate model; column 2 - sGFAP-based model (using *z*-scores of sGFAP concentration); column 3 - Combination of metabolomics and sGFAP (multivariate model; Supplementary Fig. 5). For comparison, profiles of non-transitioning individuals are presented in the SI (RRMS Supplementary Figs. 2, 3; SPMS Supplementary Fig. 4).

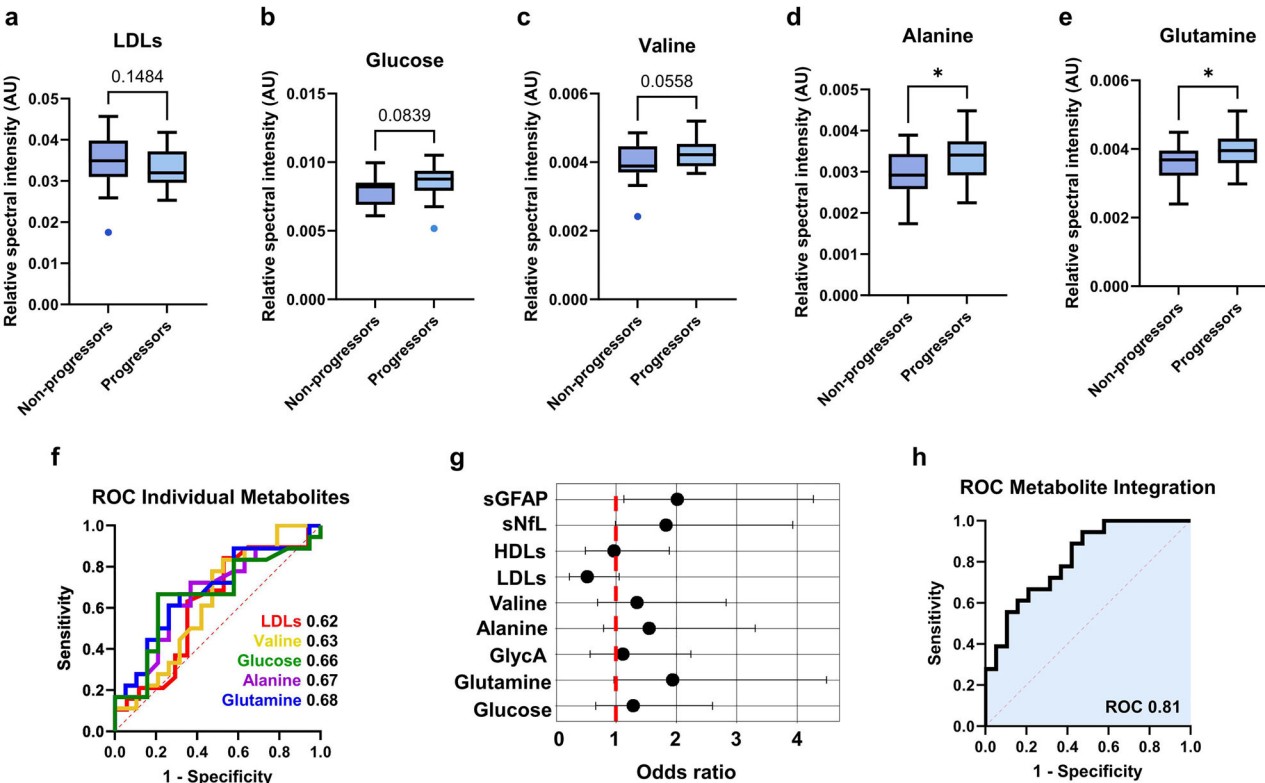

**Fig. 2 | Combining individual serum metabolites substantially improves prediction of MS progression.** Baseline levels of the most predictor metabolites for progressors and non-progressors (*$p < 0.05$): **a** LDLs; $p = 0.148$, **b** glucose; $p = 0.084$, **c** valine; $p = 0.056$, **d** alanine; $p = 0.020$, **e** glutamine; $p = 0.024$. **f** ROC curves utilising the baseline concentrations of the five metabolites discriminating between progressors and non-progressors. **g** Odds ratios (log values of biomarker concentrations; given for progressors/non-progressors) for all potential NMR-identified biomarkers and sGFAP and sNfL $z$-scores. **h** ROC curve incorporating the most discriminatory metabolites - LDLs, glucose, valine, alanine, and glutamine - achieved an AUC of 0.81 (95% CI: 0.68–0.95, DeLong $p = 0.001$), using baseline concentrations to distinguish between progressors and non-progressors. Unpaired two-sided t-tests were used for comparisons in panels **a–e**, with FDR correction applied to variables selected by VIP scores from OPLS-DA models. Sample sizes were $N = 18$ progressors and $N = 19$ non-progressors.

LDLs, alanine, glutamine, valine, GlycA, glucose) could differentiate between future progressors and non-progressors, irrespective of disease stage. We observed subtle yet significant alterations in the levels of these potential biomarkers in baseline samples matched for EDSS scores, highlighting their ability to distinguish between progressors and non-progressors (Fig. 2a–e). Additionally, ROC curve analysis indicated that most metabolites demonstrated a distinct ability to assess disease progression (Fig. 2f). The AUCs for each metabolite were as follows: HDLs (AUC 0.54 (95% CI: 0.34-0.73), t-test $p = 0.336$), GlycA (AUC 0.56 (95% CI: 0.37-0.73), t-test $p = 0.393$), LDLs (AUC 0.62 (95% CI: 0.43-0.81), t-test $p = 0.148$, Fig. 2a, f), glucose (AUC 0.66 (95% CI: 0.47-0.84), t-test $p = 0.084$, Fig. 2b, f), valine (AUC 0.63 (95% CI: 0.45-0.81), t-test $p = 0.056$, Fig. 2c, f), alanine (AUC 0.67 (95% CI: 0.49-0.84), t-test $p = 0.020$, Fig. 2d, f), and glutamine (AUC 0.68 (95% CI: 0.50-0.86), t-test $p = 0.024$, Fig. 2e, f).

The odds ratios (ORs; Fig. 2g) for baseline biomarker levels indicated moderate predictive capacity but, with the exception of glutamine, remained lower than those of sGFAP $z$-scores (OR 2.02, 95% CI: 1.13–4.27) and sNfL $z$-scores (OR 1.83, 95% CI: 1.00–3.93). Specifically, LDLs were associated with an OR of 0.52 (95% CI: 0.22–1.04), valine 1.35 (95% CI: 0.70–2.83), alanine 1.55 (95% CI: 0.79–3.30), glutamine 1.94 (95% CI: 0.96–4.49), and glucose 1.29 (95% CI: 0.66–2.60), while GlycA (OR 1.11 (95% CI: 0.47–2.24)) and HDLs (OR 0.97 (95% CI: 0.49–1.88)) exhibited minimal predictive capacity. Combining the identified predictor metabolites - LDLs, glutamine, alanine, valine, and glucose - each associated with a relatively high AUC and discriminatory OR, into a unified model significantly improved differentiation between future progressors and non-progressors, achieving an AUC of 0.81 (95% CI: 0.68–0.95, DeLong $p = 0.001$, Fig. 2h). This model accurately classified 78% of progressors (14/18) and 63% of non-progressors (12/19) based on baseline serum metabolomic profiles.

We then further validated whether the baseline concentration of the identified discriminatory metabolites could predict progression. To achieve this, we conducted Kaplan-Meier analysis, using metabolite baseline $z$-scored levels to define abundance (positive = high, negative = low), with an event defined as an EDSS increase ≥1.5 (Fig. 3). We compared these results to sGFAP and sNfL for reference. The Kaplan–Meier curves suggested a trend toward a higher risk of EDSS increase among individuals with lower baseline lipoprotein levels ($p = 0.146$), with 5-year event estimates of 63% (95% CI: 30–80%) for low and 33% (95% CI: 10–51%) for high baseline lipoproteins. Similarly, participants with higher alanine levels at baseline were more likely to experience an EDSS score increase ≥1.5 over the follow-up ($p = 0.027$, Fig. 3b), with 58% (95% CI: 27–73%) of individuals with high and 26% (95% CI: 4–44%) with low baseline alanine levels projected to progress in the next five years. Similarly, 56% (95% CI: 26%-73%) of participants with high baseline glutamine levels and 28% (95% CI: 4–42%) with low baseline levels were more likely to progress ($p = 0.052$, Fig. 3c). A similar trend was observed for valine, where 61% (95% CI: 31–78%) of participants with high baseline valine concentrations and 32% (95% CI: 7–49%) with low concentrations were likely to progress within five years ($p = 0.163$, Fig. 3d). In contrast, glucose did not show any predictive power ($p = 0.336$, Fig. 3e), with 53% (95% CI: 27–70%) of individuals with high baseline glucose concentrations and 36% (95% CI: 9–54%) with low concentrations at greater risk of progression. Similarly, the GlycA signal ($p = 0.441$, Fig. 3f) did not show predictive value for, with 55% (95% CI: 26–71%) of participants with high baseline GlycA levels and 37% (95% CI: 8%-55%) with low levels more likely to progress.

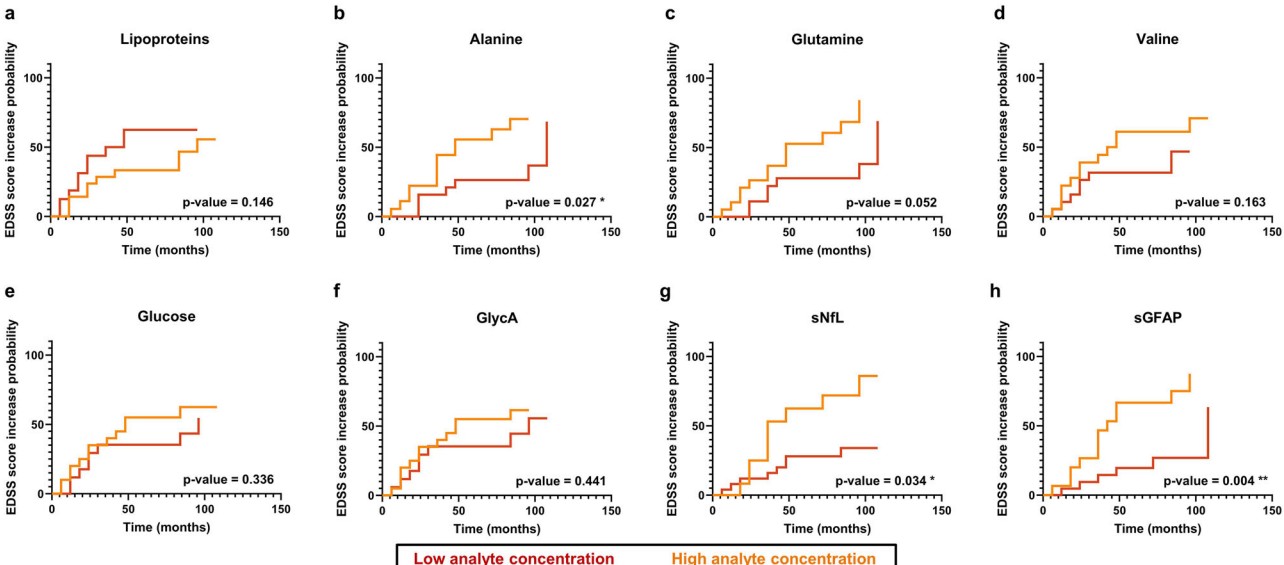

**Fig. 3 | Prediction of an increase in EDSS score based on the baseline levels of serum metabolites (HDLs, LDLs, alanine, glutamine, valine, glucose and GlycA), sGFAP and sNfL.** Kaplan-Meier curves illustrate the predictive capacity of the most discriminatory analytes. The event was defined as an EDSS score increase by ≥1.5 units. Patients were stratified based on their baseline metabolite concentration, with low (negative $z$-scores) concentration depicted in red and high (positive $z$-scores) concentration depicted in yellow. **a** Lipoproteins - the trends for both LDLs and HDLs were similar, **b** alanine, **c** glutamine, **d** valine, **e** glucose, **f** GlycA, **g** sNfL, **h** sGFAP. Log-rank (Mantel-Cox) tests were used to compare groups, with FDR correction applied to variables selected by VIP scores from OPLS-DA models. Sample sizes were $N = 18$ progressors and $N = 19$ non-progressors.

In comparison, sGFAP and sNfL concentrations revealed that individuals with higher levels were significantly more likely to progress. sNfL demonstrated similar predictive power to alanine ($p = 0.034$, Fig. 3g) with 68% (95% CI: 26–87%) of individuals with high and 28% (95% CI: 8–44%) with low baseline concentration predicted to progress. In contrast, sGFAP substantially correlated with disease progression ($p = 0.004$, Fig. 3h), as 69% (95% CI: 34%-85%) of individuals with high and 22% (95% CI: 3–38%) with low sGFAP baseline concentration were identified as progressors.

### Integration of sGFAP and NMR-metabolites enhances prediction of disease progression

Having previously established a correlation between sGFAP and sNfL concentration levels and MS progression in the SMSC cohort[16], our objective was to assess whether incorporating the NMR-derived biomarkers could enhance overall prediction accuracy. The Kaplan-Meier analysis for sNfL yielded results comparable to those for lipoprotein signals and alanine ($p = 0.034$, Fig. 3g), whereas the predictive power of sGFAP was significantly stronger ($p = 0.004$, Fig. 3h), consistent with previous studies[16]. Logistic regression comparing progressors and non-progressors showed that both sNfL (AUC 0.68 (95% CI: 0.52–0.84), $t$-test $p = 0.063$, Fig. 4a, b) and sGFAP $z$-scores (AUC 0.72 (95% CI: 0.55–0.88), $t$-test $p = 0.020$, Fig. 4a, c) demonstrated notable baseline discriminatory capacity.

We further investigated the potential benefits of integrating sGFAP and sNfL with serum metabolites to improve the differentiation between future progressors and non-progressors. Incorporating the five previously identified metabolites with the highest discriminatory capacity – LDLs, alanine, glutamine, valine, and glucose – alongside sNfL and sGFAP $z$-scores, significantly enhanced the predictive performance. Specifically, the AUC values increased to 0.87 (95% CI: 0.76–0.98, DeLong $p = 0.0002$) for sNfL and 0.91 (95% CI: 0.83–0.99, DeLong $p < 0.0001$) for sGFAP (Fig. 4e, f), indicating a strong capacity to predict progression. These findings were further corroborated using a RF algorithm, which yielded an average ROC AUC of 0.86 and an accuracy of 76.2% for the integration with sGFAP, confirming the diagnostic potential of this biomarker constellation (Supplementary Fig. 7a). Including the less discriminatory metabolites, GlycA and HDL signals, did not enhance model performance, likely due to their limited predictive value and potential collinearity with metabolites already incorporated in the model. Pearson correlation analysis (Fig. 4d) demonstrated a moderate positive correlation between sGFAP and most serum metabolites, excluding lipoproteins, which showed a negative correlation, and GlycA, which showed no correlation. Although correlations between sNfL and metabolites were moderate, integrating sNfL with the five key predictor metabolites - LDLs, alanine, glutamine, valine, and glucose - enabled the correct classification of 13 out of 18 (72%) progressors and 15 out of 19 (78%) non-progressors. Further improvement was achieved by incorporating sGFAP levels, which, when combined with these five metabolites, increased classification accuracy to 15 out of 18 (83%) progressors and 17 out of 19 (89%) non-progressors based on baseline metabolite levels.

### Independent cohort validation confirms the metabolomic findings

To assess the reproducibility and generalisability of our findings, we performed an external validation using an independent cohort, a subset of the Oxford METabolomics cohort (Supplementary Table 6). Progressors ($N = 19$) and non-progressors ($N = 24$) were selected based on extreme ambulatory trajectories, with groups matched for baseline EDSS scores ($p = 0.11$).

Univariate and ROC curve analyses demonstrated that specific serum metabolites exhibited trends consistent with the Basel 'extreme phenotypes' cohort, with significantly decreased LDLs levels in progressing individuals (AUC 0.71 (95% CI: 0.55–0.87), $t$-test $p = 0.027$, Fig. 5a, g) and, conversely, elevated glucose (AUC 0.70 (95% CI: 0.54–0.86), $t$-test $p = 0.025$, Fig. 5b, g), valine (AUC 0.59 (95% CI: 0.41–0.77), $t$-test $p = 0.235$, Fig. 5c, g), alanine (AUC 0.72 (95% CI: 0.56–0.89), $t$-test $p = 0.061$, Fig. 5d, g), and glutamine (AUC 0.72 (95% CI: 0.57–0.88), $t$-test $p = 0.015$, Fig. 5e, g) in progressing MS individuals. Multiple logistic regression incorporating these metabolites further reinforced their association with MS progression, yielding an ROC AUC of 0.85 (95% CI: 0.71–0.96; DeLong $p = 0.0001$, Fig. 5h, grey ROC), surpassing the Basel model and highlighting their potential as predictive biomarkers. Although not all metabolites reached statistical significance in the validation cohort, the consistent direction of change and the strong performance of the multivariable model indicate that predictive capacity arises from their combined contribution rather than reliance on any single marker.

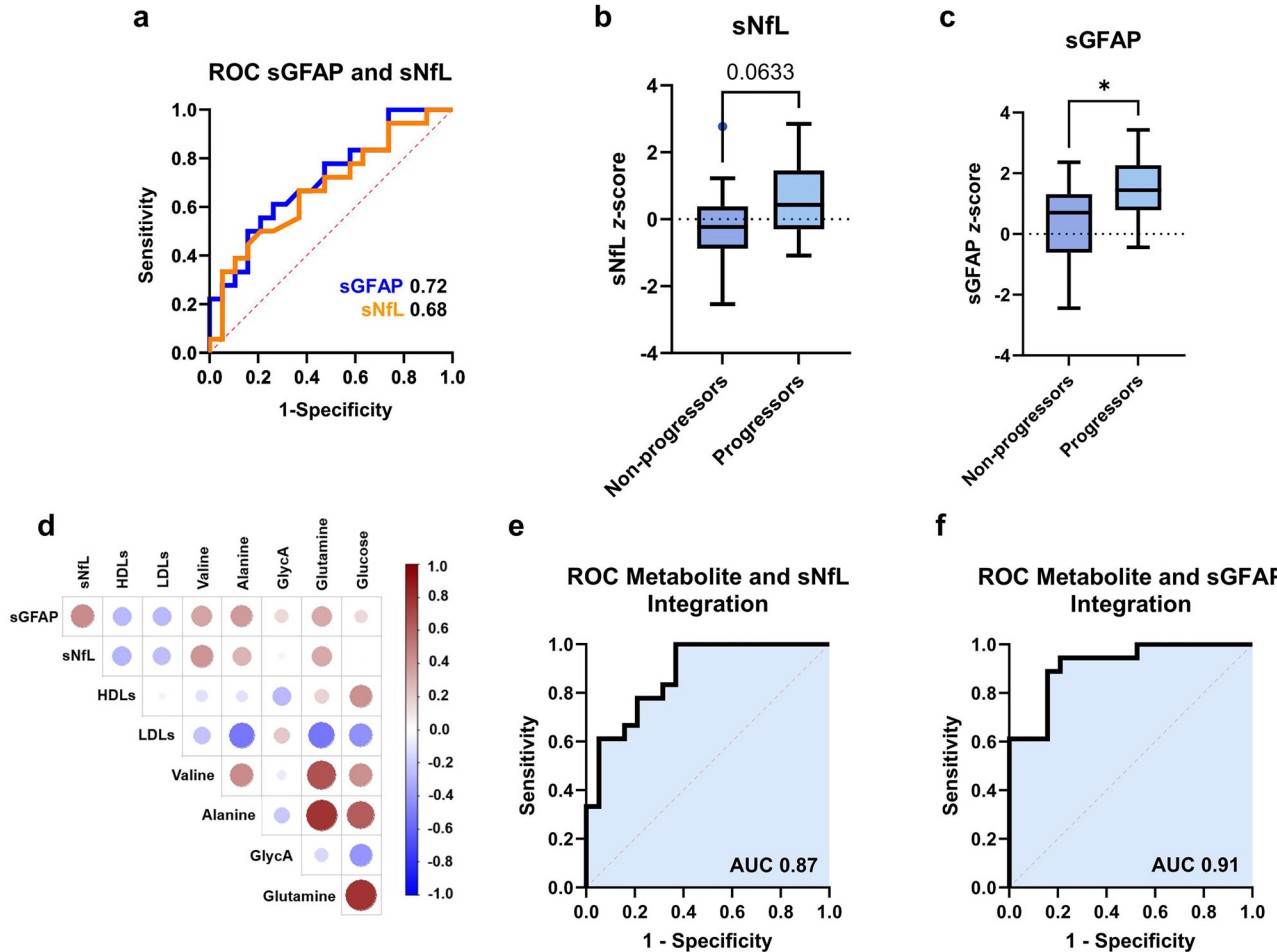

**Fig. 4 | Improved classification of future progressors and non-progressors was achieved through the combination of five metabolites (LDLs, glutamine, valine, alanine, glucose) with sGFAP or sNfL. a** ROC curves utilising baseline sGFAP and sNfL z-scores to discriminate between progressors and non-progressors. **b, c** t-test results for baseline levels of sNfL and sGFAP z-scores for progressors and non-progressors; **b** sNfL; $p = 0.063$; **c** sGFAP; $p = 0.020$. **d** Correlation analysis of sGFAP, sNfL, and serum metabolites. **e** ROC curve for the integration of five serum metabolites (LDLs, alanine, glutamine, valine, glucose) and sNfL z-scores. **f** ROC curve for the integration of five serum metabolites and sGFAP z-scores. Unpaired two-sided t-tests were used in **b, c**, and correlation in panel d was assessed using Pearson's r. Sample sizes were $N = 18$ progressors and $N = 19$ non-progressors.

In contrast to the Basel cohort, sNfL z-scored levels were not significantly different between progressors and non-progressors ($p = 0.667$; Fig. 5f) and did not demonstrate predictive utility for disease progression (AUC 0.52, 95% CI: 0.34–0.70), Fig. 5g). Despite the lack of significance, sNfL levels were elevated in progressors but exhibited substantial data variability (Fig. 5f). This variability, unlike in the Basel cohort, may reflect the shorter follow-up period, potentially limiting the ability to capture long-term progression patterns, or greater heterogeneity in disease course within the Oxford cohort, both of which would contribute to the reduced predictive effect observed. However, although sNfL alone showed limited predictive performance, its integration with metabolite data significantly improved the model (ROC AUC = 0.89, 95% CI: 0.82–0.99; DeLong $p < 0.0001$, Fig. 5h). This finding supports the central premise of the study that combining complementary serum biomarkers across molecular layers enhances predictive accuracy and provides a more robust framework for assessing disease trajectories.

## Discussion

In a rigorously defined extreme phenotype MS cohort of Swiss individuals, we identified distinct serum metabolite patterns associated with disease progression. Metabolites, sNfL, and particularly sGFAP correlated with both RRMS-to-SPMS transition and PIRA-defined progression, suggesting that baseline biomarkers can predict disease progression independently of

disease stage and inflammatory activity. Although the discovery cohort was enriched for patients at the extremes of the disease spectrum, facilitating detection of robust signals, the metabolome remained stable apart from transitioning individuals, supporting translational relevance. Notably, several metabolomic signals were replicated in an independent validation cohort with shorter follow-up and in a Finnish cohort of milder progressors, extending these findings beyond extreme phenotypes[49].

Despite evidence of metabolic differences between RRMS and SPMS, the molecular mechanisms underlying this transition remain poorly understood[34–36,39]. While the categorical distinction between RRMS and SPMS remains in clinical use, growing evidence supports a continuum of progression from disease onset, highlighting the need for reliable biomarkers to address this diagnostic gap. Our data corroborate the concept of this continuum, as they demonstrate that metabolic signatures are stable within both stages despite temporal variability, yet can detect transitions more sensitively. This is particularly relevant given that EDSS-based staging is driven by gradual functional decline, is relatively insensitive to smouldering neurodegenerative changes, and typically lags behind the onset of its underlying molecular pathology. Additionally, integrating key NMR-identified metabolites with sGFAP[13–16] suggests that a composite biomarker panel could enhance clinical assessments and reduce misclassification risks in atypical cases[4,7,50]. Given the ambiguity in RRMS-SPMS classification[1,4–6], this integration across metabolomic and protein biomarkers within a multi-

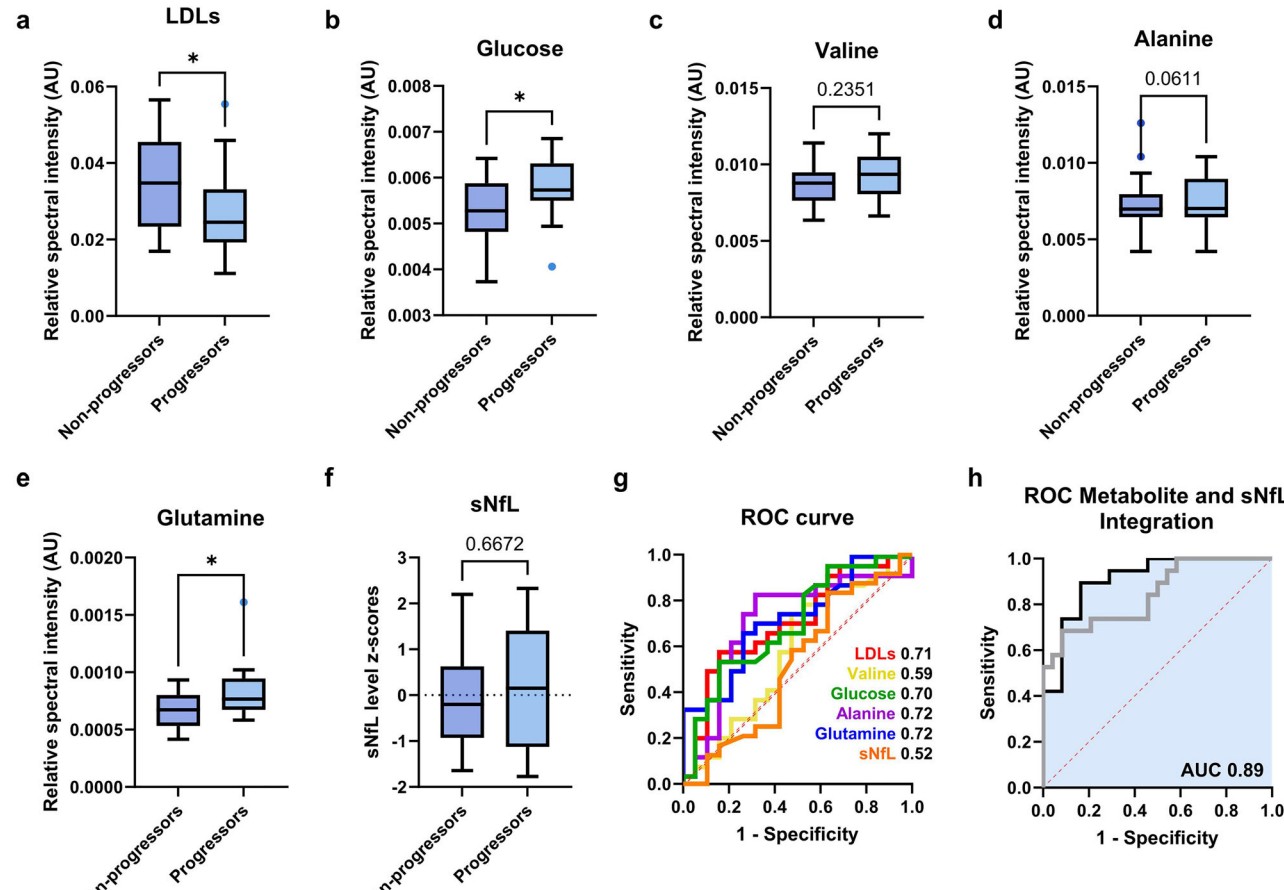

**Fig. 5 | Validation of serum metabolite-based prediction of disease progression in an independent cohort.** Baseline levels of the most predictor metabolites in progressors and non-progressors: **a** LDLs; $p = 0.027$, **b** glucose; $p = 0.025$, **c** valine; $p = 0.235$, **d** alanine; $p = 0.061$, **e** glutamine; $p = 0.015$, **f** sNfL; $p = 0.667$. **g** ROC curve analysis using baseline concentrations of these analytes to discriminate between progressors and non-progressors. **h** A model incorporating the most discriminatory metabolites - LDLs, glucose, valine, alanine, and glutamine - achieved an AUC of 0.85 (95% CI: 0.71–0.96; DeLong $p = 0.0001$, grey ROC). When combined with sNfL z-scores, the AUC increased to 0.89 (95% CI: 0.82–0.99; DeLong $p < 0.0001$), further improving the ability to distinguish between progressors and non-progressors based on baseline concentrations. Unpaired two-sided t-tests were used for panels **a–f**, with FDR correction for multiple metabolite comparisons. Sample sizes were $N = 19$ progressors and $N = 24$ non-progressors.

omics framework provides an orthogonal prognostic tool that complements existing measures, captures information beyond PIRA events, and maintains a strong association with disease progression.

The association between inflammatory diseases and altered lipid metabolism, has been well-documented[51–53]. Building on this, our previous work reported that lipoprotein NMR profiles can effectively distinguish RRMS from SPMS[34–36]. Previous studies have further shown that elevations in very-low-density lipoprotein (VLDLs) sub-fractions, alongside overall dyslipidaemia, are associated with increased lesion load and disability progression in MS patients[54]. Here, we demonstrate that LDLs and HDLs may also serve as progression biomarkers, as their serum levels reflect both disease staging and future worsening.

While the link between peripheral glucose levels and MS progression requires further investigation, the subtle glucose increase in progressors may indicate increased metabolic activity or dysregulation in active MS[55]. Similarly, the increased need for rapid energy sources could be attributed to elevated energy demands in the brain, driven by inflammation and demyelination as shown in experimental studies[56]. Previous studies have demonstrated that glucose levels can discriminate between RRMS and SPMS[34], and it has also been hypothesised that there is a link between disease severity and insulin resistance[57].

Amino acids play a pivotal role in the central nervous system, functioning as neurotransmitters, neuromodulators, and energy regulators[58]. Neurodegeneration and neuroinflammation disrupt amino acid homeostasis, driven by metabolic alterations, oxidative stress, and gastrointestinal protein malabsorption[59–61]. We propose that the elevated serum glutamine in progressing MS reflects increased glutamine efflux across the blood–brain barrier, likely driven by impaired neurotransmitter synthesis[62–64]. Mechanistically, impaired mitochondrial function in neurons and astrocytes may reduce the conversion of glutamine to glutamate and GABA, forcing excess glutamine export into the circulation and disturbing excitatory–inhibitory balance[65]. We further observed elevated serum valine in progressing MS individuals. While BCAA dysregulation has not been previously linked to MS[38], altered BCAA levels have been reported in Parkinson's disease and amyotrophic lateral sclerosis[66,67]. Given the role of excessive glutamate release in neurodegeneration[68,69], BCAA imbalance may contribute to MS pathophysiology through impaired nitrogen shuttling and altered mTOR signalling, which in turn exacerbate neuroinflammatory cascades[65]. Additionally, evidence suggests a connection between neurodegeneration and disrupted alanine metabolism[66,70], with one study reporting increased serum alanine in SPMS and its potential involvement in disease progression[71].

Expanding on prior research linking MS progression to sGFAP[13,15,16,41] our study integrates established biomarkers (sGFAP, sNfL) with serum metabolites, offering deeper insights. By combining complementary molecular layers, the approach outperforms single-biomarker models, as shown in both the discovery and validation cohorts. Given the early onset of neurodegeneration[1,4,5,72], this biomarker

combination enables individualised prognostic assessment and early intervention[73]. The feasibility, non-invasiveness, and cost-effectiveness of measuring sGFAP and NMR-metabolites in blood samples make this approach a promising tool for personalised MS management, particularly as NMR metabolomics delivers multiplexed readouts that complement targeted assays such as Simoa and strengthen integration with clinical and MRI-based evaluations.

## Potential confounds

Identified metabolites showed no significant correlations with age, disease duration, sex, or pharmacotherapy. Although baseline EDSS and lesion volume tended to be higher in progressors, these differences were not statistically significant and are unlikely to account for the observed metabolic associations. NMR metabolite levels did not differ between treated and untreated individuals, though the small subgroup sizes and variation in DMT use (including four untreated progressors) limited assessment of treatment effects.

## Limitations

The main limitation of this study is the relatively small sample size; however, this is partially offset by the long-term, consistent follow-up and the selection of extreme MS phenotypes, which provide a valuable framework for future research. The limited sample size constrains cross-validation using baseline samples alone; however, identified metabolites were validated through rigorous machine learning models (OPLS-DA, RF), and even successfully validated in an independent cohort. Additionally, in a separate MS cohort, elevated serum glutamine and glucose were also associated with mild progression ($\Delta$EDSS/year $\geq 0.5$), supporting broader clinical applicability[49]. Although sNfL exhibited lower discriminative power in the external validation cohort, this may reflect differences in follow-up duration and disease heterogeneity, potentially affecting predictive accuracy. These findings highlight the promise of this approach while underscoring the need for further validation in a larger cohort incorporating sGFAP and sNfL measurements.

## Conclusion

In an extreme-phenotype MS cohort with long-term follow-up, this study demonstrates that integrating serum NMR-detected metabolites with sGFAP and sNfL improves prognostication of subclinical progression. The metabolomic findings further support the concept of progression as a continuum in MS pathogenesis, using an orthogonal approach to ELISA-based techniques. By leveraging multiple variables across distinct omics layers, this integrative strategy outperforms single-component models, offers a more comprehensive view of disease pathology, and enables a more individualised, data-driven approach to disease management. Furthermore, findings from independent cohorts suggest the potential for translating these biomarkers into larger populations for broader disease progression prediction and monitoring.

## Data availability

Clinical data analysed in this study are fully anonymised but cannot be made openly available due to restrictions associated with participant consent and data-sharing agreements, which limit distribution to controlled access. Anonymised clinical data not included within this article may be made available to qualified researchers upon reasonable request to the corresponding author, subject to institutional approval and completion of a data use agreement governing confidentiality, permitted analyses, and reporting conditions. Data may be used only for the approved research purpose and may not be redistributed or used for participant re-identification. Requests for access should be directed to the corresponding author, Prof. Daniel Anthony (daniel.anthony@pharm.ox.ac.uk). Requests will be acknowledged within 5 working days and reviewed within 4 weeks, subject to institutional review processes. The metabolomics dataset generated as part of this study has been deposited in the University of Oxford Research Archive (ORA) and is publicly available under the DOI: 10.5287/ora-

xo5mbmxo1. Source data underlying the main figures are provided as Supplementary Data 1.

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

## Acknowledgements

We thank all MS patients participating in this study and the NMR Research Facility (Department of Chemistry, University of Oxford) for use of equipment and advice. This research was supported in part by the Aqua-Synapse EU framework (2022–2026), funded by the European Union's Horizon 2020 Research and Innovation programme under the Marie Sklodowska-Curie Grant Agreement No. 101086453 awarded to D.C.A. F.P. is funded by a Dorothy Hodgkin Early Career Fellowship in Chemistry in association with Somerville College. T.K. is funded by an EPSRC Doctoral Training Partnership (EP/W524311/1), EPSRC talent and skills funding, and Numares AG (Am Biopark 9, 93053 Regensburg-Graß, Germany).

## Author contributions

J.K., D.C.A., F.P., and D.L. conceived and designed the study. T.K. and E.W. performed the experiments. T.K., D.E.R.S., and W.X. analysed the data. J.K., D.L., E.W., J.O., and A.M.M. oversaw the Swiss MS cohort. M.S., L.S., J.P., G.L., T.Y., F.P., and D.C.A. oversaw the Oxford MS cohort. J.K., D.C.A., F.P., D.L., and E.W. provided critical feedback and guidance for experimental and analysis design. T.K. drafted the manuscript with input from all authors. All authors reviewed and approved the manuscript.

## Competing interests

The authors declare no competing interest.

## Additional information

**Tereza Kacerova** ⓘ**¹, Eline Willemse²,³,⁴,⁵, Johanna Oechtering²,³,⁴, Daniel E. Radford-Smith⁶, Wenzheng Xiong⁶, Megan Sealey⁶, Luisa Saldana⁷, Aleksandra Maleska Maceski²,³,⁴, Tianrong Yeo** ⓘ**⁸,⁹,¹⁰,¹¹, Gabriele DeLuca⁷, Jacqueline Palace⁷, David Leppert²,³,⁴, Jens Kuhle** ⓘ**²,³,⁴, Daniel C. Anthony** ⓘ**⁶ ✉ & Fay Probert¹**

¹Chemistry Research Laboratory, Department of Chemistry, University of Oxford, Oxford, UK. ²Department of Neurology, University Hospital and University of Basel, Basel, Switzerland. ³Multiple Sclerosis Centre, Departments of Biomedicine and Clinical Research, University Hospital and University of Basel, Basel, Switzerland. ⁴Research Center for Clinical Neuroimmunology and Neuroscience Basel, University Hospital and University of Basel, Basel, Switzerland. ⁵Department of Clinical Research, University Hospital Basel, University of Basel, Basel, Switzerland. ⁶Department of Pharmacology, University of Oxford, Oxford, UK. ⁷Nuffield Department of Clinical Neurosciences, John Radcliffe Hospital, University of Oxford, Oxford, UK. ⁸Department of Neurology, National Neuroscience Institute , Singapore, Singapore. ⁹Duke-NUS Medical School, Singapore, Singapore. ¹⁰Lee Kong Chian School of Medicine (Nanyang Technological University), Singapore, Singapore. ¹¹A*STAR Institute of Molecular and Cell Biology, Singapore, Singapore. ✉e-mail: daniel.anthony@pharm.ox.ac.uk

