## [Transparent Peer Review file · Communications Medicine]

Serum GFAP and NfL Augment a Metabolomics-Driven Strategy for Long-Term Prediction of Multiple Sclerosis Progression

Corresponding Author: Professor Daniel Anthony

Version 0:

Reviewer comments:

Reviewer #1

(Remarks to the Author)

In this manuscript the authors use NMR based metabolomics and sNfL and GFAP measures to try and separate RRMS and SPMS and potentially predict disease progression. Given our understanding of the pathology of the disease – the focus on trying to “stage” the disease seems unreasonable – especially with a method like metabolomics. We know that both inflammation and neurodegeneration are happening throughout the disease – though to varying extents. Overall the small sample size and unclear statistical methods in addition to inconsistencies between treatment of the cohorts make it unclear if any of these results are meaningful.

Major concerns

1. Small sample sizes – with unclear comparison of demographic characteristics in the various subsets.
2. Why not have similar definitions for progression in the two cohorts ?
3. It is strange that 4 patients in the progressor group were untreated even at end of follow up time period
4. Also a majority of baseline characteristics are trending in the wrong direction in the progressor group (EDSS, lesion volume)
5. Most of the metabolites have low AUCs on their own
6. Not clear why they show sGFAP + metabolomics prediction in one cohort and then switch over to sNfL + metabolomics cohort in validation

Reviewer #2

(Remarks to the Author)

This manuscript by Kacerova et al. presents a rigorous, well-executed investigation into the utility of serum NMR-based metabolomics, alongside glial fibrillary acidic protein (GFAP) and neurofilament light chain (NfL), for predicting long-term disease progression in multiple sclerosis (MS). The study leverages an "extreme phenotype" cohort from the Swiss Multiple Sclerosis Cohort Study (SMSC), with validation in an independent Oxford cohort, employing a comprehensive suite of statistical and machine learning techniques.

The work addresses an important unmet clinical need and represents a significant contribution to the growing body of literature focused on blood-based biomarkers for neurodegenerative disease monitoring. The integrative analytical approach, longitudinal design, and external validation are notable strengths.

However, some areas warrant clarification or further development. These issues do not undermine the overall integrity or importance of the work, but addressing them would enhance the manuscript's clarity, translational relevance, and generalisability.

1. The manuscript presents strong predictive performance (e.g., AUC > 0.9 when combining metabolites with sGFAP or sNfL), but the relatively modest sample size (particularly in the training cohort) raises the possibility of model overfitting.

Please provide additional information on the number of features used in the final logistic regression and random forest models, and justify the feature-to-sample ratio.

2. The proposed integrative biomarker panel is promising; however, the practical steps needed to bring such an approach into clinical use are not well developed. Elaborate on the clinical feasibility of implementing NMR metabolomics as a routine tool in MS monitoring. How does it compare, in terms of cost, scalability, and required expertise, to existing platforms such as Simoa for GFAP and NfL?

3. While the study identifies several metabolites associated with disease progression (e.g., glutamine, alanine, lipoproteins), the discussion on underlying pathophysiology is brief and, in places, speculative. Strengthen the discussion by referencing existing metabolic or neuropathological studies that might support the biological relevance of these metabolites in MS progression.

4. Both the training and validation cohorts are derived from "extreme phenotype" populations, which improves signal detection but may limit generalisability. Address this limitation explicitly in the discussion. Consider outlining a plan for validating this model in a broader, more heterogeneous MS population.

Reviewer #3

(Remarks to the Author)

Overview: This study evaluates the predictive potential of serum nuclear magnetic resonance (NMR)-based metabolomics, individually and in combination with well-established biomarkers of neuroinflammation (serum glial fibrillary acidic protein, sGFAP) and axonal damage (neurofilament light chain, sNfL), in an extreme-phenotype subset of the Swiss Multiple Sclerosis Cohort (SMSC).

Comments: This is a timely manuscript as paraclinical tests that improve prediction of disability are needed. While the authors use novel methods to evaluate metabolomic profiles as predictors, the consistency of these profiles between the tested cohorts is unclear.

Strengths:

1. Authors use a well characterized extreme phenotype cohort to test predictive biomarkers as well as a strong validation cohort

Weaknesses:

1. There are limited number of progressors and non-progressors, possibly impacting reproducibility of biomarkers.

2. Identified biomarkers did not correlate with the transition point to SPMS. It would be intuitive to think that, if metabolomic profiles were related to progression, higher or steeper profile changes would relate to time-to-transition

3. It is unclear which metabolite(s) impact predictability. Lower baseline lipoprotein was associated with a higher likelihood of transition to SPMS when compared with any of the other metabolites. However, the authors report that the "metabolomics-based" model had higher agreement with the clinical diagnosis than the serum GFAP-based model and integration of the two marker types further increased agreement.

4. The authors report on page 13, lines 316-319, that persons with lower baseline sLDLs were more likely to experience significant increase in EDSS appears incorrect since the reported p-value is not significant ($p=0.146$).

5. How do the authors account for the variability in disability prediction with the different metabolites between the two cohorts? For example, while the authors report that "specific serum metabolites exhibited trends consistent with the Basel extreme phenotypes cohort", some of the specific markers were not significant.

Version 1:

Reviewer comments:

Reviewer #2

(Remarks to the Author)

I previously raised a number of methodological and interpretive points regarding model overfitting, cohort representativeness, biological interpretation of identified metabolites, and clinical implementation feasibility. I am pleased to note that these concerns have now been satisfactorily addressed in the revised manuscript.

Reviewer #3

(Remarks to the Author)

Response to Reviewer Comments

Manuscript Title: *Serum GFAP and NfL Augment a Metabolomics-Driven Strategy for Long-Term Prediction of Multiple Sclerosis Progression*

Manuscript ID: COMMSMED-25-0473 R1 Kacerova *et al.*

Corresponding Author: Anthony D. C.

Address: Department of Pharmacology, University of Oxford, Mansfield Road, Oxford OX1 3QT, UK

Email: daniel.anthony@pharm.ox.ac.uk

Tel: +44(0)1865 281136

The reviewers' comments are shown in black, followed by our responses in *blue*. Newly added or revised text is **highlighted in yellow**. Page and line numbers refer to the revised (annotated) manuscript. In the edited manuscript and Supplementary Information, all added or modified text is **underlined in red** for clarity.

Reviewer 1**Remarks to the Author:**

In this manuscript the authors use NMR based metabolomics and sNfL and GFAP measures to try and separate RRMS and SPMS and potentially predict disease progression. Given our understanding of the pathology of the disease – the focus on trying to “stage” the disease seems unreasonable – especially with a method like metabolomics. We know that both inflammation and neurodegeneration are happening throughout the disease – though to varying extents. Overall, the small sample size and unclear statistical methods in addition to inconsistencies between treatment of the cohorts make it unclear if any of these results are meaningful.

Response

We thank the reviewer for their detailed comments and for carefully considering our manuscript. We fully acknowledge the challenges inherent to biomarker studies in MS, particularly regarding cohort size, disease heterogeneity, and the limitations of stage-based classifications. While our study initially presented results in relation to RRMS versus SPMS, we recognise and agree with the reviewer’s key point that the clinical distinction between these stages is increasingly viewed as arbitrary and retrospective, given that both inflammatory and neurodegenerative processes occur throughout the disease course. Indeed, recent consensus suggests that MS progression is best conceptualised as a continuum from disease onset, rather than a discrete transition between RRMS and SPMS.

Our analyses, however, were not designed to reinforce this dichotomy but rather to test the broader concept that integrating serum metabolomics with established biomarkers of astrocytic and neuroaxonal injury (sGFAP and sNfL) can predict progression itself. By selecting individuals with extreme phenotypes and long-term follow-up, we aimed to define the molecular “signature of a progressor,” which is arguably more clinically relevant than RR/SPMS staging. Our data corroborate the continuum concept, demonstrating that metabolic signatures remain stable within both stages despite temporal variability, yet can sensitively detect transitions. This is particularly important given that EDSS-based staging reflects gradual functional decline, is relatively insensitive to smouldering neurodegenerative changes, and often lags behind the onset of underlying molecular pathology.

To reflect this point more clearly, we have revised the Abstract (pp. 2, ll. 46-58), Introduction (pp. 5, ll. 108-112), Discussion (pp. 20, ll. 460-466), and Conclusion (pp. 22, ll. 528-534). The revised text now emphasises that: (1) our results demonstrate the ability to predict progression independent of relapse activity and independent of stage, (2) integrating metabolomics with sGFAP and sNfL consistently outperforms any single biomarker or stage-based model, and (3) our findings support the view that molecular biomarkers of progression represent a more reliable and clinically actionable framework for future MS management than conventional RR/SPMS staging.

We would also like to draw attention to the fact that both Reviewer 2 and Reviewer 3, independently, considered the statistical methodology appropriate, the study design robust, and the findings translationally relevant. Reviewer 2 highlighted the rigorous feature selection, appropriate feature-to-sample ratio, and external validation as major strengths, while Reviewer 3 noted the robustness of the extreme phenotype design and the reproducibility of biomarkers across cohorts. Taken together, these assessments provide important reassurance that our results are both methodologically sound and clinically meaningful.

In this context, we position our results not as definitive biomarkers ready for immediate clinical adoption, but as proof-of-concept evidence that multi-omics integration outperforms single-marker approaches and offers a promising strategy for future biomarker development in MS. We believe this balanced framing appropriately acknowledges the reviewer's concerns while also reflecting the positive consensus from the broader reviewer panel.

Major concerns

Comment 1:

Small sample sizes – with unclear comparison of demographic characteristics in the various subsets.

Response

We thank the reviewer for this comment. We acknowledge that the overall sample sizes, particularly in the discovery cohort, are modest. This reflects our deliberate strategy to select individuals representing extreme phenotypes with consistent long-term follow-up (5–8 years) from the larger Swiss MS Cohort (SMSC), as assembling such a well-defined and clinically homogeneous cohort is challenging given the inherent heterogeneity of MS. This design increases signal detection and reduces the risk of clinical misclassification. We have now provided a clearer comparison of demographic and clinical characteristics across all relevant subsets, as shown in Supplementary eTables 1-6. These, alongside Table 1, show that progressors and non-progressors were well matched for sex, age, BMI, and baseline EDSS, with no significant differences except for the expected outcomes related to progression (e.g., EDSS at last sample, frequency of PIRA events). The same applies to the Oxford validation cohort, where groups were matched for baseline EDSS and age. In addition, the subsets used for the staging models (RRMS and SPMS) were also derived from the same main Extreme Phenotypes (part of SMSC) cohort; their demographics were well matched, and we have clarified these details in the Supplementary Tables. We have expanded the Results (pp. 8, ll. 198-201) to clarify cohort selection and description.

Comment 2:

Why not have similar definitions for progression in the two cohorts?

Response:

We thank the reviewer for this thoughtful point. The progression definitions differed because of the distinct designs and available follow-up data in the two cohorts. In the SMSC discovery cohort, progression was defined using PIRA (progression independent of relapse activity) events, supported by 5–8 years of longitudinal EDSS data. This allowed us to apply a stringent, sustained definition of disability worsening.

In the Oxford validation cohort, follow-up data were collected during the COVID-19 pandemic, which limited the scope of standardised assessments. To ensure comparability, we therefore focused on extreme ambulatory trajectories, selecting individuals with clear and clinically meaningful differences: progressors who developed marked walking impairment (≤ 500 m walking distance) and non-progressors who remained fully ambulatory over two years. These criteria were established in close collaboration with the neurologists and clinicians responsible for the cohort, ensuring that they faithfully reflected the principle of extreme phenotype selection applied in Basel.

Thus, while the operational definitions necessarily differed, both approaches captured unambiguous disability worsening in line with the cohorts' available data and clinical context. We have clarified this distinction in the revised Methods Section 2.1 (pp. 6, ll. 128-129).

Comment 3:

It is strange that 4 patients in the progressor group were untreated even at end of follow up time period

Response:

We thank the reviewer for noting this. A small proportion of progressors were indeed untreated at the last follow-up, reflecting real-world treatment heterogeneity in MS. As described in our 'potential confounds section (4.1)', NMR metabolite levels did not differ between DMT-treated and untreated individuals; however, we acknowledge that the small subgroup sizes and treatment variability limited our ability to formally assess treatment effects. We have clarified this point further in the revised Discussion – Potential confounds (pp. 22, ll. 513-515).

Comment 4:

Also a majority of baseline characteristics are trending in the wrong direction in the progressor group (EDSS, lesion volume)

Response:

We thank the reviewer for this observation. While baseline EDSS and T2 lesion volume values were numerically higher in the progressor group, these differences were not statistically significant (Table 1). This trend likely reflects the clinical reality that individuals at greater risk of progression often present with

slightly higher baseline disability or lesion burden, despite careful matching for EDSS at study entry. Importantly, progressors and non-progressors remained well matched overall for demographic and clinical variables, and the progression-associated metabolites identified in our study did not correlate with baseline EDSS or lesion volume. We have included this as a comment in the Discussion – Potential confounds section (pp. 20, ll. 510-513).

Comment 5:

Most of the metabolites have low AUCs on their own

Response

We thank the reviewer for this important comment. We fully agree that most individual metabolites exhibited only modest discriminatory performance, with relatively low AUC values when considered in isolation. This is consistent with previous metabolomics studies in MS and other neurological diseases, where single analytes rarely provide sufficient predictive power to serve as reliable biomarkers. The central aim of our study was therefore not to propose individual metabolites as standalone markers, but to assess whether combining metabolite signatures with established protein biomarkers of neuroaxonal damage (sNfL) and astrocytic activation (sGFAP) could improve prediction of disease progression.

To address this, we selected discriminatory metabolites from OPLS-DA models (using the full binned dataset, with results validated by Random Forest analysis) and carried forward those above the inflection point of the VIP score distribution for subsequent logistic regression. While the predictive value of individual metabolites was modest, the combined metabolite panel consistently outperformed any single analyte. Importantly, integration of this metabolite panel with sGFAP and sNfL further enhanced discrimination (achieving AUCs > 0.9), underscoring the value of an integrative, multi-omics framework.

We believe this reflects the biological complexity of MS progression, which cannot be captured by a single molecular pathway. Instead, complementary information across distinct molecular layers - metabolomic, astrocytic, and neuroaxonal - provides a more comprehensive picture of disease biology and yields greater predictive accuracy. We have clarified this rationale in the revised Discussion (pp. 20, ll. 462-466) and Conclusion (pp. 22, ll. 530-534), emphasising that the strength of our approach lies in integrating multiple omics domains rather than relying on any one metabolite or marker, and supporting the broader concept that future biomarker development in MS should prioritise multi-omics integration.

Comment 6:

Not clear why they show sGFAP + metabolomics prediction in one cohort and then switch over to sNfL + metabolomics cohort in validation

Response

We thank the reviewer for raising this important point. The difference reflects both sample availability and cohort characteristics. In the SMSC discovery cohort, serum samples with reliable sGFAP and sNfL measurements were available and well suited for integration with metabolomics. In contrast, sGFAP was not

consistently available in the Oxford cohort. Instead, sNfL was available in this cohort and therefore used as the complementary biomarker layer for validation.

We would like to emphasise that our primary goal was not to validate a specific fixed panel of markers (e.g., sGFAP + metabolites), but rather to test the broader concept that combining metabolomic signatures with established fluid biomarkers enhances predictive performance. In this study, that meant integrating untargeted metabolomics with targeted proteomic biomarkers of astrocytic (sGFAP) or neuroaxonal (sNfL) injury. The use of sNfL in the Oxford cohort therefore illustrates the adaptability of this integrative, multi-omics strategy and demonstrates that the approach is not restricted to one biomarker pairing. Importantly, both cohorts together validate the principle that predictive accuracy improves when complementary molecular layers are combined, rather than relying on any single marker. This supports the future development of multi-omics biomarker panels in MS, aimed at improving individualised risk stratification and clinical decision-making.

We have clarified this point in the revised Results (pp. 19, ll. 444-448) and Discussion (pp. 21, ll. 501-503).

Reviewer 2

Remarks to the Author:

This manuscript by Kacerova et al. presents a rigorous, well-executed investigation into the utility of serum NMR-based metabolomics, alongside glial fibrillary acidic protein (GFAP) and neurofilament light chain (NfL), for predicting long-term disease progression in multiple sclerosis (MS). The study leverages an "extreme phenotype" cohort from the Swiss Multiple Sclerosis Cohort Study (SMSC), with validation in an independent Oxford cohort, employing a comprehensive suite of statistical and machine learning techniques.

The work addresses an important unmet clinical need and represents a significant contribution to the growing body of literature focused on blood-based biomarkers for neurodegenerative disease monitoring. The integrative analytical approach, longitudinal design, and external validation are notable strengths.

However, some areas warrant clarification or further development. These issues do not undermine the overall integrity or importance of the work, but addressing them would enhance the manuscript's clarity, translational relevance, and generalisability.

Response

We thank Reviewer 2 for their thoughtful and constructive feedback, and for the time taken to evaluate our manuscript. Their comments have helped us to clarify key points and strengthen the paper. In response, we have specified the number of predictors in the final models, expanded the discussion on the clinical feasibility of NMR metabolomics, added mechanistic context for the main metabolites, and addressed the limitations of using extreme phenotype cohorts. We believe these revisions have improved the clarity of the manuscript and reinforced its overall message.

Comment 1:

The manuscript presents strong predictive performance (e.g., AUC > 0.9 when combining metabolites with sGFAP or sNfL), but the relatively modest sample size (particularly in the training cohort) raises the possibility of model overfitting. Please provide additional information on the number of features used in the final logistic regression and random forest models, and justify the feature-to-sample ratio.

Response

We thank the reviewer for this important point. To minimise the risk of overfitting in light of the modest sample size, we applied strict feature selection procedures and limited the number of predictors in the final models. NMR-derived metabolites were first evaluated using OPLS-DA across the full binned dataset, and only variables with high variable importance in projection (VIP) scores above the inflection point of the distribution were retained. This ensured that only robust discriminatory features were carried forward. Final logistic regression models incorporated five NMR-derived metabolites (LDLs, alanine, glutamine, valine, and glucose) together with sGFAP and/or sNfL z-scores, resulting in a total of 5-6 predictors. With 37 individuals (18 progressors, 19 non-progressors) in the SMSC discovery cohort, this corresponded to a feature-to-sample ratio of approximately 1:6-1:7, which is within accepted limits when predictors are preselected through cross-validated methods. Random Forest models were restricted to the same preselected features (≤ 10 variables), with overfitting further mitigated by bootstrap aggregation, out-of-bag validation, and repeated 10-fold cross-validation (100 iterations with 500 trees), which consistently produced stable performance metrics. Importantly, external validation in the independent Oxford cohort ($N = 43$) reproduced these associations, achieving AUCs of 0.85 for metabolites alone and 0.89 for metabolites combined with sNfL. This replication provides strong evidence that the predictive signal is robust rather than an artefact of model overfitting. We have clarified the number of predictors in the revised Methods section (pp. 8, ll. 191-192).

Comment 2:

The proposed integrative biomarker panel is promising; however, the practical steps needed to bring such an approach into clinical use are not well developed. Elaborate on the clinical feasibility of implementing NMR metabolomics as a routine tool in MS monitoring. How does it compare, in terms of cost, scalability, and required expertise, to existing platforms such as Simoa for GFAP and NfL?

Response

We thank the reviewer for raising this important point regarding clinical translation. We agree that feasibility is central to evaluating the potential of NMR metabolomics in MS monitoring. Over the past decade, NMR-based lipoprotein and metabolite profiling has advanced substantially towards clinical use. Automated NMR platforms such as Bruker IVDr (Clinical Research | Bruker) and numares AXINON® (Numares | Innovation. For Precision Diagnostics.) are already CE-marked and employed in accredited European laboratories for standardised lipoprotein testing. These systems require minimal preparation (~10

minutes per sample, simple buffer addition), and provide fully automated data acquisition and quantification, demonstrating that large-scale, standardised NMR-based metabolomics can be incorporated into routine workflows.

Compared with Simoa, which offers highly sensitive and relatively low-cost assays for sGFAP and sNfL once infrastructure is in place, NMR metabolomics has the advantage of delivering a broad metabolic and lipoprotein profile from a single experiment with comparable turnaround times. While the upfront capital cost of high-field NMR instrumentation is greater than that of immunoassay platforms, the per-sample running cost (€30–50) is competitive, particularly as multiple analytes are quantified simultaneously without additional reagents.

In terms of scalability and expertise, Simoa assays are straightforward to operate in clinical laboratories, whereas NMR has traditionally required more specialised infrastructure. However, advances in automated processing pipelines have reduced operator dependency, and the development of benchtop NMR systems (60–80 MHz) offers a promising, more affordable route for broader implementation. Although less sensitive than high-field systems, these benchtop instruments are increasingly able to quantify key metabolites and lipoproteins reproducibly, and may eventually support decentralised or point-of-care applications.

In summary, while Simoa currently represents the lower barrier to adoption, NMR metabolomics is becoming increasingly feasible, offering multiplexed outputs, regulatory acceptance, and future scalability. Unlike immunoassays, NMR quantifies a wide range of metabolites and lipoproteins simultaneously from a single sample, making it highly efficient for routine use. Importantly, we view NMR and Simoa as complementary rather than competing approaches, with the greatest translational potential likely to come from their integration. We reflected this idea in the updated Discussion (pp. 21, ll. 504-508).

Comment 3:

While the study identifies several metabolites associated with disease progression (e.g., glutamine, alanine, lipoproteins), the discussion on underlying pathophysiology is brief and, in places, speculative. Strengthen the discussion by referencing existing metabolic or neuropathological studies that might support the biological relevance of these metabolites in MS progression.

Response

We thank the reviewer for this constructive suggestion. We agree that providing additional mechanistic context strengthens the biological plausibility of our findings. In the revised Discussion, we have expanded on the potential roles of the key metabolites, noting prior evidence linking VLDL dyslipidaemia with lesion load and disability (pp. 20, ll. 475-477), impaired mitochondrial conversion of glutamine affecting neurotransmitter balance (pp. 21, ll. 489-492), and BCAA imbalance contributing to neuroinflammatory cascades (pp. 21, ll. 495-496). Due to word count constraints, we have kept these additions focused and concise. Importantly, the primary aim of this manuscript is to highlight the diagnostic potential of serum

biomarkers, with the central message being that integrating multiple omics layers can improve prediction of MS progression. We view this as an important step towards developing personalised diagnostic approaches, while more detailed mechanistic studies will be necessary to fully delineate the biochemical underpinnings of MS progression.

Comment 4:

Both the training and validation cohorts are derived from "extreme phenotype" populations, which improves signal detection but may limit generalisability. Address this limitation explicitly in the discussion. Consider outlining a plan for validating this model in a broader, more heterogeneous MS population.

Response

We thank the reviewer for this important comment. We agree that the use of extreme phenotype cohorts can enhance detection of strong biomarker signals but may limit direct generalisability. Importantly, our discovery cohort was very carefully selected by expert clinicians, and we demonstrated that, apart from individuals in transition, the metabolome remained remarkably stable, supporting its translational potential. In addition, in a completely independent Finnish cohort that included much milder progressors with shorter follow-up, we recently observed that two of the reported markers, glutamine and glucose, were again among the top VIPs, further supporting the robustness of our findings. We emphasise that the present study was exploratory in nature, and we fully agree with the reviewer that application of these models in larger and more diverse MS populations will be essential for clinical translation. We have now expanded the Discussion to address this point (pp. 19-20, ll. 454-458).

Reviewer 3**Remarks to the Author**

Overview: This study evaluates the predictive potential of serum nuclear magnetic resonance (NMR)-based metabolomics, individually and in combination with well-established biomarkers of neuroinflammation (serum glial fibrillary acidic protein, sGFAP) and axonal damage (neurofilament light chain, sNfL), in an extreme-phenotype subset of the Swiss Multiple Sclerosis Cohort (SMSC).

Comments: This is a timely manuscript as paraclinical tests that improve prediction of disability are needed. While the authors use novel methods to evaluate metabolomic profiles as predictors, the consistency of these profiles between the tested cohorts is unclear.

Response

We thank the reviewer for their thoughtful and constructive comments. We agree that improving the consistency and reproducibility of biomarker findings across cohorts is a key challenge in the field, and their feedback has helped us to clarify several important aspects of our study design, analysis, and interpretation.

We believe that the revisions made in response to these comments have strengthened the manuscript, improved transparency, and better highlighted the translational significance of our findings.

Strengths:

1. Authors use a well characterized extreme phenotype cohort to test predictive biomarkers as well as a strong validation cohort

Weaknesses:**Comment 1:**

There are limited number of progressors and non-progressors, possibly impacting reproducibility of biomarkers.

Response

We thank the reviewer for this important point. Although the SMSC discovery cohort was modest in size, as noted in both the Results and Limitations, individuals were carefully selected as extreme phenotypes with 5–8 years of longitudinal follow-up, reducing misclassification and enhancing the signal-to-noise ratio. Within this dataset, the same metabolites (LDLs, alanine, glutamine, valine, glucose) consistently emerged across statistical approaches, including two independent machine-learning algorithms, supporting internal robustness. These biomarkers replicated in an independent Oxford MS cohort (N = 43), where they differentiated progressors from non-progressors with an AUC of 0.85 for metabolites alone and 0.89 when combined with sNfL. Glutamine and glucose were further validated in a Finnish MS cohort (Turku; PMID 40248672), where they distinguished stable from progressing patients over one year. Importantly, these alterations, reflective of disease stage, are consistent with our prior Oxford cohorts (PMID: 32709911; 31693024; 25253748) and with larger MS and neurodegeneration studies, underscoring their biological relevance and translational potential. We have clarified this in the revised Discussion (pp. 19, ll. 454-458) and Limitations (pp. 21, ll. 521-523).

Comment 2:

Identified biomarkers did not correlate with the transition point to SPMS. It would be intuitive to think that, if metabolomic profiles were related to progression, higher or steeper profile changes would relate to time-to-transition

Response

We thank the reviewer for raising this important point. It might be expected that metabolomic trajectories would map directly onto the clinical transition from RRMS to SPMS. However, the absence of such a correlation likely reflects limitations of clinical staging rather than a lack of biological relevance. The diagnosis of SPMS is retrospective, based largely on gradual EDSS-defined decline, which is relatively insensitive to subtle neurodegenerative changes and often lags behind molecular pathology.

Our findings suggest that metabolomic alterations may precede, accompany, or follow the clinically defined transition, indicating that these biomarkers capture progression at a molecular level not fully aligned with the arbitrary RRMS/SPMS dichotomy. This interpretation is consistent with evidence that progression independent of relapse activity (PIRA) can occur during the RR phase, and that disability worsening does not always coincide with formal stage transitions. We therefore view the metabolomic signals as complementary not only to clinical staging but especially to sGFAP and sNfL, with the combination providing a more complete picture of disease stage. This was evident in Figure 1, where they contributed to the disease-staging model, but became particularly clear when examining prediction at a more granular level - independent of the RRMS/SPMS distinction - by focusing directly on progression itself.

We have clarified this point in the revised Discussion (pp. 20, ll. 460-464), noting that the divergence between metabolomic trajectories and time-to-transition underscores the value of molecular biomarkers in capturing progression dynamics beyond those detected by EDSS-based staging.

Comment 3:

It is unclear which metabolite(s) impact predictability. Lower baseline lipoprotein was associated with a higher likelihood of transition to SPMS when compared with any of the other metabolites. However, the authors report that the “metabolomics-based” model had higher agreement with the clinical diagnosis than the serum GFAP-based model and integration of the two marker types further increased agreement.

Response

We thank the reviewer for this comment and have clarified the text accordingly. In univariate analyses, only lipoproteins (LDLs and HDLs) were significantly associated with time-to-transition ($p = 0.045$), whereas glutamine, alanine, valine, and glucose did not reach significance for this endpoint, although they showed consistent trends. While individual metabolites may not be strongly predictive on their own, the multivariate “metabolomics-based” model combined several discriminatory metabolites identified by OPLS-DA and Random Forest, capturing a broader metabolic signature that reached high accuracy in both staging and predicting progression. This explains why the multivariate model achieved higher accuracy than sGFAP alone. Although sGFAP is a well-established marker of progression, it represents only one molecular dimension; our findings demonstrate that integrative multivariable models, incorporating complementary metabolic and protein biomarkers, provide a more powerful and comprehensive readout of disease biology. These clarifications have been added to the revised Results (pp. 13, ll. 294-296).

Comment 4:

The authors report on page 13 [15], lines 316-319 [338-341], that persons with lower baseline LDLs were more likely to experience significant increase in EDSS appears incorrect since the reported p-value is not significant ($p=0.146$).

Response

We thank the reviewer for this observation. They are correct that the association did not reach statistical significance ($p = 0.146$), and our original wording could be misinterpreted as implying significance. We have revised the text to reflect this more accurately, describing it as a trend (pp. 15, ll. 338-341).

Comment 5:

How do the authors account for the variability in disability prediction with the different metabolites between the two cohorts? For example, while the authors report that “specific serum metabolites exhibited trends consistent with the Basel extreme phenotypes cohort”, some of the specific markers were not significant.

Response

We thank the reviewer for raising this important point. We agree that not all metabolites reached statistical significance in the Oxford validation cohort, although they generally exhibited trends consistent with the Basel discovery cohort. Several factors are likely to contribute to this variability. First, the Oxford cohort was larger but also more heterogeneous, with a broader spectrum of patients, a shorter follow-up period, and less stringent inclusion criteria compared to the carefully selected “extreme phenotypes” in the Basel cohort. The Basel design, by focusing on patients at the ends of the disease spectrum, reduced clinical heterogeneity and misclassification, thereby amplifying metabolic signals associated with progression. In contrast, the Oxford cohort reflects a more representative clinical population, in which subtle metabolic differences are diluted by greater biological and clinical variability. This inevitably attenuates the statistical strength of individual metabolites, especially when assessed independently in more heterogeneous clinical cohorts.

Despite these differences, the direction of change for the key metabolites (LDLs, glucose, glutamine, alanine, valine) was consistent across both cohorts, supporting their biological reproducibility. This concordance in trends is critical, as it suggests that the same underlying metabolic pathways are involved, even if effect sizes and p -values vary depending on cohort design. We note that such variability is expected in translational research, where biomarker performance often differs across populations due to differences in demographics, disease duration, treatment exposure, and follow-up intervals. Rather than undermining the findings, these observations highlight the importance of validating candidate biomarkers in multiple independent cohorts to ensure their robustness and generalisability.

Importantly, even though not all metabolites were individually significant in the Oxford cohort, the multivariate metabolite panel reproduced strong predictive performance ($AUC = 0.85$), and its integration with sNfL further improved accuracy ($AUC = 0.89$). This indicates that predictive capacity arises from the constellation of metabolites acting together rather than from reliance on any single marker. Such integrative models, which combine complementary layers of metabolic and protein biomarkers, are more resilient to variability between cohorts and better reflect the complex biology of disease progression.

We have clarified these points in the revised Results section (pp. 19, ll. 434-437). By validating both individual metabolites and composite biomarker panels across distinct populations (Basel, Oxford, and consistencies were observed also with the Turku cohort), our study strengthens confidence in their reproducibility and potential clinical utility.